# Microbiologically Influenced Corrosion of Q235 Carbon Steel by *Ectothiorhodospira* sp.

**DOI:** 10.3390/ijerph192215416

**Published:** 2022-11-21

**Authors:** Hong Qi, Yingsi Wang, Jin Feng, Ruqun Peng, Qingshan Shi, Xiaobao Xie

**Affiliations:** Key Laboratory of Agricultural Microbiomics and Precision Application (MARA), Guangdong Provincial Key Laboratory of Microbial Culture Collection and Application, Key Laboratory of Agricultural Microbiome (MARA), State Key Laboratory of Applied Microbiology Southern China, Institute of Microbiology, Guangdong Academy of Sciences, Guangzhou 510070, China

**Keywords:** corrosion, steel, pitting, electrochemistry, *Ectothiorhodospira* sp.

## Abstract

The biological sulfur cycle is closely related to iron corrosion in the natural environment. The effect of the sulfur-oxidising bacterium *Ectothiorhodospira* sp., named PHS-Q, on the metal corrosion behaviour rarely has been investigated. In this study, the corrosion mechanism of Q235 carbon steel in a PHS-Q-inoculated medium is discussed via the characterization of the morphology and the composition of the corrosion products, the measurement of local corrosion and the investigation of its electrochemical behaviour. The results suggested that, initially, PHS-Q assimilates sulfate to produce H_2_S directly or indirectly in the medium without sulfide. H_2_S reacts with Fe^2+^ to form an inert film on the coupon surface. Then, in localised areas, bacteria adhere to the reaction product and use the oxidation of FeS as a hydrogen donor. This process leads to a large cathode and a small anode, which incurs pitting corrosion. Consequently, the effect of PHS-Q on carbon steel corrosion behaviour is crucial in an anaerobic environment.

## 1. Introduction

Metals inevitably undergo corrosion in the natural environment. One of the most ubiquitous corrosion factors in the natural environment is the presence of microbes. The economic loss incurred by microbiologically influenced corrosion (MIC) accounts for 20% of the total corrosion amount [1,2]. Current studies reported that microorganisms causing metal corrosion can be roughly divided into the following categories: sulfate-reducing bacteria (SRB), sulfur-oxidising bacteria (SOB), iron-oxidising bacteria (IOB), iron-reducing bacteria (IRB), acid-producing bacteria (APB) and slime producing bacteria (SPB) [3,4,5,6,7,8]. The corrosion of metals by SRB, among the aforementioned bacteria, has been investigated considerably and there has been quite a lot of mature theories. The biological sulfur cycle in nature is well known to be closely related to iron corrosion [9,10]. Nevertheless, SOB is closely related to the sulfur cycle, and its effect on the corrosion of metals is not investigated much. Corresponding research mainly focuses on aerobic SOB, by which metals are corroded by produced sulfuric acid [11,12]. However, the corrosion influenced by anaerobic SOB, which are widely present in an anaerobic environment such as sediments, has not been widely investigated. Especially, facultative autotrophic SOB exist, which can oxidise sulfide to sulfate under photoautotrophic conditions to be used as the hydrogen donor and, thus, to fix carbon dioxide. In addition, sulfate can be used as an electron acceptor under photoheterotrophic conditions, where sulfate is reduced to sulfide. Sulfates and sulfides are common in nature. Accompanied with the presence of such SOB, the transformation of the sulfur valence may transform a friendly environment into a corrosive environment for metals. For these reasons, the effect of such SOB on the metal corrosion behaviour has attracted our research attention. It is imperative to conduct relevant research to determine the corrosion mechanisms. The research results also could serve as compelling evidence to verify the close relationship between the sulfur cycle and iron corrosion.

In this study, a strain belonging to facultative autotrophic SOB was obtained, i.e., *Ectothiorhodospira* sp. In addition, the effect of *Ectothiorhodospira* sp. on the corrosion behaviour of carbon steel was investigated. The composition and morphology of the corrosion products and substrate were analysed to verify if this bacterial species would cause pitting corrosion on carbon steel. Then, the mechanism of pitting corrosion on the metal was investigated by electrochemical analysis. The results of this study demonstrated that bacteria that can simultaneously oxidise and reduce sulfur-containing inorganic minerals in an anaerobic environment are corrosive to carbon steel, confirming the close relationship between the sulfur cycle and iron corrosion. This study also could provide deep research insights into the understanding of MIC.

## 2. Materials and Methods

### 2.1. Bacterial Identification and Cultivation

The photosynthetic bacterium *Ectothiorhodospira* sp. used here was named PHS-Q and examined by the sequence analysis of 16S rRNA gene, gram staining and morphology. The bacteria were cultivated at 37 °C at light intensities of 3000 to 4000 lux under anaerobic conditions (N_2_:H_2_:CO_2_ (80:10:10)) in the medium containing (grams per liter in distilled water): NH_4_Cl 1.0 g, K_2_HPO_4_ 0.2 g, MgSO_4_∙7H_2_O 0.2 g, CH_3_COONa 3.0 g, NaHCO_3_ 1.0 g, Yeast extract 0.1 g and inorganic salt solution 1 mL (pH 7.5).

### 2.2. Metal Material Preparation and Immersion Corrosion Test

The Q235 carbon steel was chosen as the research metal. The Q235 coupons were cut square-shaped (40 × 15 × 2 mm^3^), abraded with 400 and 800-grit sandpapers sequentially, and then rinsed with acetone, ethanol and distilled water sequentially. The coupons were dry-heated for sterilization (160 °C, 2 h). Three replicate coupons and 250 mL cultivation medium were added to each sterile bottle. The bacterial inoculation of the experimental group was 10%. The corrosion test was conducted at 37 °C at light intensities of 3000 to 4000 lux under anaerobic conditions (N_2_:H_2_:CO_2_ (80:10:10)) in the cultivation medium (pH: 7.5). The test was repeated thrice with newly polished coupons and fresh cultivation medium for reproducibility. The period of the immersion corrosion test was 14 days.

### 2.3. Morphological, Elemental Composition and Pit Depth Characteriazitions

The morphology of the bacterium was observed by a Hitachi H-7650 (Hitachi, Tokyo, Japan) transmission electron microscopy (TEM) (LaB_6_ filament, 80 kV, point resolution: 0.2 nm). The morphology and element distribution of the coupon surfaces were detected by a Mira LMS (TESCAN, Brno-Kohoutovice, CZ) scanning electron microscope (SEM) with Oxford Xplore 30 (Oxford Instruments plc, Abingdon, UK) energy-dispersive X-ray spectrometer (EDS). The chemical composition of the corrosion product was obtained by the Rigaku Smartlab (Rigaku, Tokyo, Japan) X-ray diffraction (XRD), the confocal Raman microscope Alpha300R (WiTech, Ulm, Germany) with a 633 nm laser transmitter and the Thermo Scientific K-Alpha Nexsa (Thermo Fisher scientific, New York, NY, USA) X-ray photoelectron spectroscopy (XPS). The pit depth was measured by an optical infinite focus microscope (IFM) VHX-7000 (Keyence, Osaka, Japan).

### 2.4. Electrochemical Measurements

The electrochemical test, mainly about electrochemical impedance spectroscopy (EIS) and potentiodynamic polarization, was performed on an electrochemical workstation CS350 (Corrtest, Wuhan, China) to characterise the electrochemical reactions and the formation of corrosion products and biofilms at the metal/biofilm interface. The electrochemical tests were performed in an anaerobic corrosion cell with a three electrodes system. The working electrode, reference electrode and counter electrode were the Q235 steel, saturated calomel electrode and platinum mesh, respectively. The cultivation medium was used as the electrolyte. The experiment group was inoculated with PHS-Q seed culture to provide an initial planktonic cell count of 10^6^ cells mL^−1^. The open circuit potential (OCP) was steady after testing for about 1 h. EIS was performed based on the OCP, and the amplitude was set as 10 mV, with frequencies ranging from 100 kHz to 10 MHz. Potentiodynamic polarization curves were obtained at a scan rate of 1 mV s^−1^. EIS of the system was tested on the initial, 1st, 2nd, 4th, 7th and 14th day, and the potentiodynamic polarization curves were tested on the 14th day. The data were analysed by CS Studio 5. Each test was duplicated using three separate cells.

## 3. Results

### 3.1. Strain Identification

The strain PHS-Q was identified as Gram-negative, motile and helical bacteria (dimensions: 2.7–3.2 μm), as shown in Figure 1. The colour of the cell suspension of the strain grown in the light was deep-red. Typically, the strain grows photoheterotrophically in the presence of light and carbon dioxide using sulfur compounds as the electron donor. Meanwhile, this strain can assimilate reduced sulfate, either when grown photoautotrophically with hydrogen as the electron donor or when grown photoheterotrophically [13,14].

**Figure 1 ijerph-19-15416-f001:**
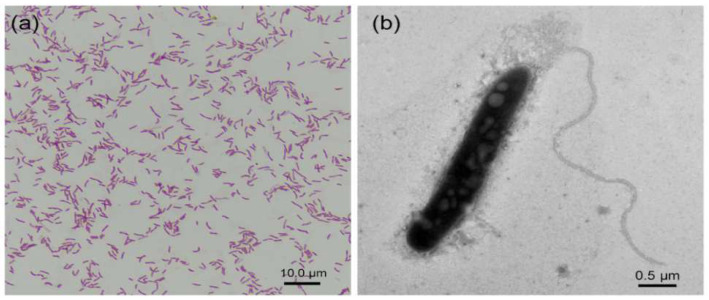
Strain identification results. (**a**) Optical microscope image of the Gram-staining cells and (**b**) TEM image of the strain.

### 3.2. Morphological, Elemental Composition and Pit Depth

Initially, polishing scratches were observed on the as-prepared Q235 coupon surface (Figure 2a). After corrosion in a sterile medium for 14 days, the coupon was uniformly corroded by the medium, and the scratches disappeared (Figure 2b). After the immersion of the as-prepared coupon in the PHS-Q-inoculated medium for 1 day, a large amount of PHS-Q cells were adsorbed on the coupon surface (Figure 2c). After incubation for 14 days, a layer of a network-like biofilm mixed with corrosion products adhered to the coupon surface, and bacteria were clearly observed in it (Figure 2d). Under the biofilm and the corrosion products, some groups of pits were spread irregularly on the coupon surface (Figure 2e,f). Pits were visible to the naked eye. The pit depth was measured by an infinite focus microscope (IFM), as shown in Figure 3. The deepest pit was located in the center of each group, and its depth reached ~5.5 μm.

The composition of the product was detected. As shown in Figure 4a, the predominance of C, Na and O in the network-like product was displayed. S and Fe were mainly distributed in the plain area of the product layer. X-ray diffraction (XRD) only can detect the signal of the substrate (Figure 4b). In the Raman spectrum of the product (Figure 4c), three typical peaks located at 214.85, 278.83 and 384.20 cm^−1^ corresponded to pyrrhotite [15,16], while the peaks centered at 676.87 cm^−1^ corresponded to Fe_3_O_4_ [17]. To further determine the chemical composition, X-ray photoelectron spectroscopy (XPS) was employed. As shown in the high-resolution O 1s spectrum in Figure 4d, three distinct peaks located at 529.6, 531.3 and 535.2 eV corresponded to Fe_3_O_4_, FeOOH and bacteria [18,19], respectively. In the high-resolution S 2p spectrum, three distinct peaks centered at 161.2, 162.4 and 164.9 eV corresponded to FeS_2_, Fe_1−x_S and S, respectively (Figure 4e) [20]. In the high-resolution N 1s spectrum, peaks at 397.2, 398.4, 399.5 and 400.5 eV corresponded to nitride, pyridine-N, protein-N and pyrrole-N, respectively (Figure 4f) [18]. In the high-resolution C 1s spectrum, four peaks located at 284.8, 286.3, 287.9 and 289.5 eV corresponded to bacteria and contamination (Figure 4g) [21]. The P 2p on the coupon surface confirmed the presence of bacteria (Figure 4h). Finally, the high-resolution Fe 2p spectrum revealed that five peaks centered at 706.6, 708.2, 710.0, 711.1 and 714.36 eV corresponded to FeS_2_, Fe_1−x_S, Fe_3_O_4_, FeOOH and FeS, respectively (Figure 4i) [22].

### 3.3. Electrochemical Measurements

Corrosion was further characterised by the electrochemical method. Figure 5a shows the variation of OCP vs. time for the sterile and PHS-Q-inoculated conditions. The OCP values of the coupon under sterile conditions increased during the incubation. The OCP values of the coupon under PHS-Q-inoculated conditions decreased within the first 4 days and kept steady until the 14th day. This drop in potential could be attributed to an increase in the dissolution kinetics of the specimen surface caused by the PHS-Q. Figure 5b shows the potentiodynamic polarization curves of the coupon, and Table 1 shows the corresponding parameters. The data show that the corrosion current densities *I_corr_* and the corrosion potential *E_corr_* of the sample in the PHS-Q-inoculated medium is less than those in the sterile media on the 14th day. The value of *E_corr_* was decreased in the PHS-Q-inoculated medium, suggesting that the presence of PHS-Q changed the corrosion mechanism. The value of *I_corr_* was also decreased in the PHS-Q-inoculated medium, indicating the uniform corrosion rate on the 14th day was inhibited. The Tafel slope is correlated with the rate of electrode reaction. A smaller value of the Tafel slope is indicative of easier electron transference [22]. For the coupon incubated in the PHS-Q-inoculated medium, the anode Tafel slope *β_a_* is smaller, and the cathodic Tafel slope *β_c_* is larger than that of the coupon incubated in a sterile medium, indicating that the kinetics of anode oxidation reactions were promoted. PHS-Q made a detrimental effect on the Q235 or the protective film formed on the coupon surface, such as pitting corrosion. On the other hand, the cathodic reactions might be correlated with the development of biofilm or corrosion products. The anodic current peak appeared at approximately −0.54 V in the PHS-Q-inoculated medium, indicating that the presence of PHS-Q could promote the passivation of the protection film but fail to start the spontaneous passivation. The corresponding potential can be defined as pitting potential *E_pit_*. The aforementioned results reveal that the Q235 carbon steel specimens undergo different corrosion processes in the presence and absence of the PHS-Q bacterium in the nutrient-rich medium. 

EIS is a non-destructive technique used to investigate electrochemical reactions at the interface between metal and corrosion products. Figure 6a–d shows the EIS data obtained from Nyquist and Bode plots. Under sterile conditions, in the Nyquist plot (Figure 6a), the diameter of the impedance loop increases with time. The respective Bode plot (Figure 6c) shows that the impedance modulus value was 3.11 on the first day. It increased rapidly on the 2nd day and ranged from 3.42 to 3.74 until the 14th day. The result reveals that the reaction resistance of the Q235 steel in the sterile medium increased with time. The Bode phase angle plot shows that the maximum angle was observed initially in the frequency range of 1 to 10 Hz. For the remaining 14 days, the maximum angle first shifted to a high frequency from the second to the fourth day. Then, it shifted to a low frequency from the seventh day. Finally, it reached the 0.01 to 0.1 Hz range. The shift of the maximum angle in the Bode plot might indicate the cyclic formation and desorption of the reaction production film. Under PHS-Q-inoculated conditions, the Nyquist plot (Figure 6b) shows that the diameter of the impedance loop increases in the first 7 days and then decreases on the 14th day. The respective Bode plot (Figure 6d) shows that the impedance modulus value increased gradually before the fourth day, ranging from 4.40 to 4.47 during the remaining days. These impedance modulus values are larger than those in the sterile medium. Combining the Bode phase angle plot, the maximum angle was observed in the frequency range of 1 to 10 Hz. A peak shift was not observed, and the peak broadened to a range of 0.1 to 10 Hz after testing for 14 days. The aforementioned results indicate that a layer of the protective film gradually formed on the coupon surface in the PHS-Q-inoculated medium from the initial day and constructed a steady protective film on the fourth day of incubation. On the 14th day, the slight decrease in impedance modulus was induced by the localised breakdown of the protective film. Moreover, the different impedance modulus values in sterile and PHS-Q-inoculated media reveal that the uniform corrosion rate of Q235 steel was slower in the PHS-Q-inoculated medium.

EIS data of the sterile and the PHS-Q-inoculated group were fitted well with the two-time constant circuit model, as shown in Figure 6e,f. Typically, this equivalent circuit can be used to represent a thin layer of product deposited on a metal surface [23,24]. In the equivalent circuit, *R_s_* stands for the solution resistance. *R_b_* and *Q_b_* stand for the resistance and the capacitance of the biofilm/corrosion product film. *R_f_* and *Q_f_* stand for the resistance and capacitance of the reaction product film in the sterile group. *R_int_* and *Q_int_* stand for the charge transfer resistance and capacitance of the film–substrate interface. The impedance of constant phase element (CPE) was used and shown below:(1)$$Z_{Q}=Y_{0}^{-1}{\left(j\omega \right)}^{-n}$$
in which *Y*_0_ is the parameter related to the capacitance. Factor *n* is CPE power. It is an adjustable parameter that lies between 0 and 1 and can be used to measure the surface inhomogeneity. The smaller the value of *n*, the larger roughness of the surface. The factor *ω* is the angular frequency. The fitting parameters are displayed in Table 2. *R_f_* + *R_int_* and *R_b_ + R_int_* were closely related to the corrosion rate: The higher the value, the lower the corrosion rate. The data show that *R_f_* and *R_b_* are much smaller than *R_int_.* Thus, the corrosion rate is decided by the value of *R_int_*. The value of *R_int_* in the PHS-Q-inoculated medium was greater than that in a sterile medium. The *R_int_* data indicated that, in the PHS-Q-inoculated medium, the films were protective and decreased the uniform corrosion rate of metals. The *Q_b_* was less than *Q_f_*. According to the definition of a capacitor, the value of a capacitor is inversely related to its thickness. Thus, the biofilm/corrosion product film formed in the PHS-Q-inoculated medium is thicker than the reaction product film formed in the sterile medium. These results are consistent with the test results of potentiodynamic polarization and the analysis of the EIS plots.

## 4. Discussion

In this study, a strain belonging to *Ectothiorhodospira* sp. [22] was obtained and identified by the sequence analysis of the 16S rRNA gene, appearance and Gram straining. This strain was named PHS-Q, a facultative autotrophic bacterium. It obtains protons mainly by the oxidation of sulfide under light conditions. In addition, it can assimilate reduced sulfate during heterotrophic photosynthesis [12,13,25,26]. The presence of bacteria can promote the bidirectional transformation of sulfur valence. The oxidation and reduction of sulfur significantly impact that of metals.

In this study, the corrosion of carbon steel Q235 was investigated. The SEM results revealed that a large number of bacteria are attached to the metal surface, gradually forming a biofilm network. A large number of cells was distributed inhomogeneously in the biofilm. The chemical composition of the corrosion products in the PHS-Q-inoculated group was detected by EDS, XRD, Raman spectrum and XPS. The results showed that the corrosion products were mainly composed of FeOOH and iron sulfide. However, hydrogen sulfide did not exist in the cultivation medium. Sulfur can only originate from the biological reduction of sulfate by PHS-Q. According to the results of electrochemical experiments, with the appearance of PHS-Q, the OCP values shifted in the negative direction revealing that the Q235 carbon steel specimens undergo different corrosion processes in the presence and absence of PHS-Q bacterium in the nutrient-rich medium. The test results of potentiodynamic polarization show that the value of *I_corr_* was decreased in the PHS-Q-inoculated medium, indicating the uniform corrosion rate was inhibited. The EIS results further reveal that, in the PHS-Q-inoculated medium, a layer of the protective film formed on the Q235 coupon surface, which decreased the uniform corrosion rate of metals. Combined with the results of IFM, under the protective film, groups of pits were distributed randomly on the metal surface, which is consistent with typically observed MIC pitting [27]. The deepest pit in the middle of the pit group reached 5.5 μm. These results revealed that *Ectothiorhodospira* sp. could change the corrosion process of the Q235 steel in the cultivation medium. It could assist in the formation of a layer of protective film on the steel surface to reduce uniform corrosion. It could also induce pitting on metals in a localised area in a short period.

According to the experimental results, combined with the metabolic characteristics of *Ectothiorhodospira* sp. and the composition of the medium, a possible mechanism for the corrosion of carbon steel by PHS-Q was proposed (Figure 7). In the medium without sulfosalt except sulfate, PHS-Q consumed the organic carbon source for photosynthetic heterotrophic metabolism. In the process, PHS-Q reduced sulfate to organic sulfide. With the decomposition of organic sulfide, H_2_S was released, permeating through the biofilm to the metal surface and forming iron sulfide with iron ions. The protective film formed on the carbon steel surface was composed of iron sulfide and a biofilm. In local areas, PHS-Q colonises directly on the iron sulfide surface. The sufficient sulfide and light initiated the pathway of photosynthetic autotrophic metabolism of PHS-Q. The sulfide formed on the surface near the PHS-Q-colonised area was gradually broken down, even exposing the metal substrate. The exposed area of the Q235 coupon was very small compared with the protective corrosion layer area. As we know, the *I_corr_* is the average corrosion current of the whole electrode plane. Thus, the value of corrosion current *I_corr_* was reduced by PHS-Q. The impedance modulus values were increased by PHS-Q. The exposed fresh metal surface continued to form the small anode and large cathode corrosion coupled with the surrounding protective film, which accelerated the corrosion rate of the metal matrix, eventually forming a pit. During bacterial adsorption and subsequent development into a biofilm, the biofilm exhibited the characteristic of being thick in the middle and thin at the edge. Thus, the pits located in the middle were deep, and those located at the edge were shallow.

## 5. Conclusions

In this study, the *Ectothiorhodospira* sp., named PHS-Q, could cause pitting corrosion of Q235 carbon steel. The corrosion behaviour and corrosion rate were characterised and analysed. Combined with the medium composition, the corrosion mechanism was preliminarily determined according to the metabolic characteristics of PHS-Q. A reasonable hypothesis of the corrosion mechanism was proposed. The conclusions are as follows:(1)The presence of the strain PHS-Q promoted the formation of a protective film on the carbon steel surface. The protective film composed of the biofilm and corrosion product (mainly the iron sulfide) slowed the uniform corrosion rate to a certain extent.(2)The PHS-Q cells attached to and oxidised the iron sulfide, which made the protective film discompose and incur pitting.

The effect of PHS-Q on Q235 corrosion should not be ignored. Meanwhile, in this study, the close relationship between the biological sulfur cycle and iron corrosion was verified.

## Figures and Tables

**Figure 2 ijerph-19-15416-f002:**
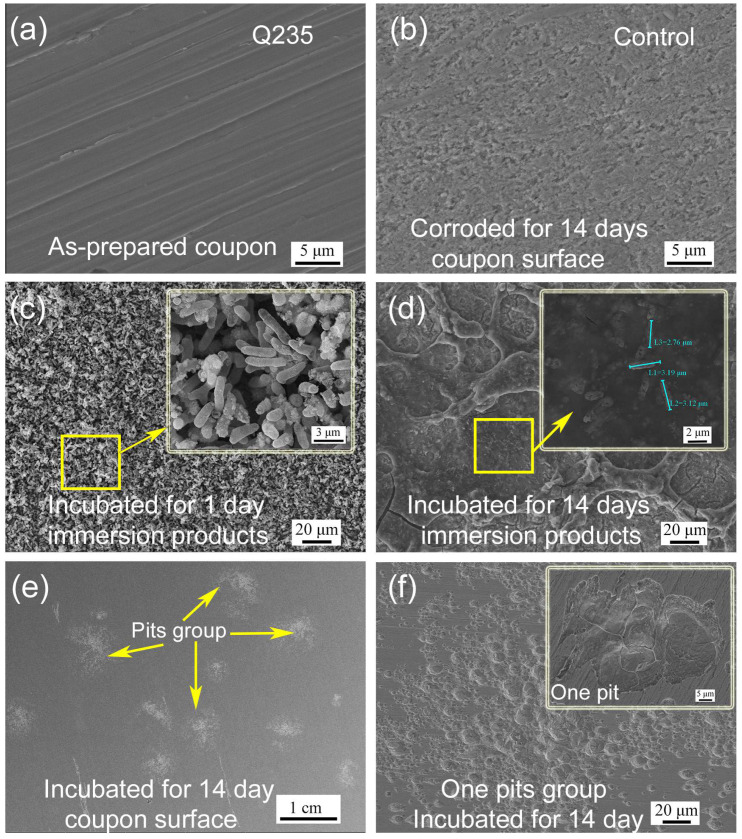
The surface morphology SEM images. (**a**) As-prepared Q235 carbon steel, (**b**) coupon corrosion in the sterile medium for 14 days, (**c**) adsorbed cells on the coupon incubated 1 day in the PHS-Q-inoculated medium, (**d**) the biofilm and products, (**e**) the distribution of pits groups on the coupon surface and (**f**) one pits group (the illustration shows the enlarged view of one pit) after 14 days incubation in the PHS-Q-inoculated medium.

**Figure 3 ijerph-19-15416-f003:**
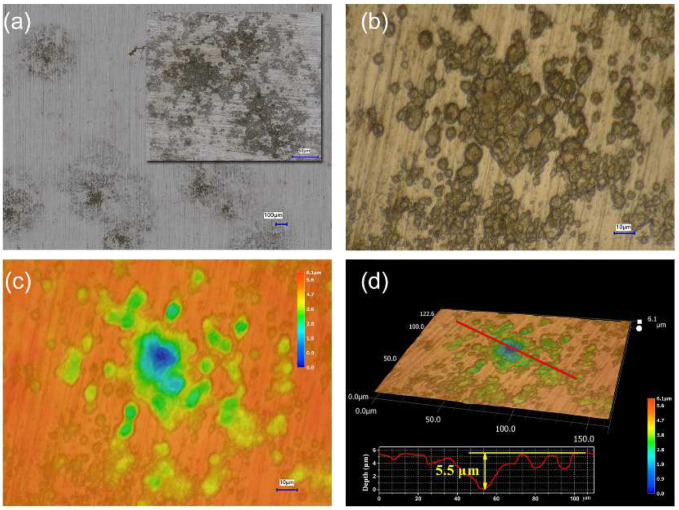
The pit depth. (**a**) Optical photographs of pits groups, (**b**) optical photograph of one pits group, (**c**) the height heat map of the pits group shown in (**b**) and (**d**) the 3D morphology and depth of pits on the coupon incubated in the PHS-Q-inoculated medium for 14 days.

**Figure 4 ijerph-19-15416-f004:**
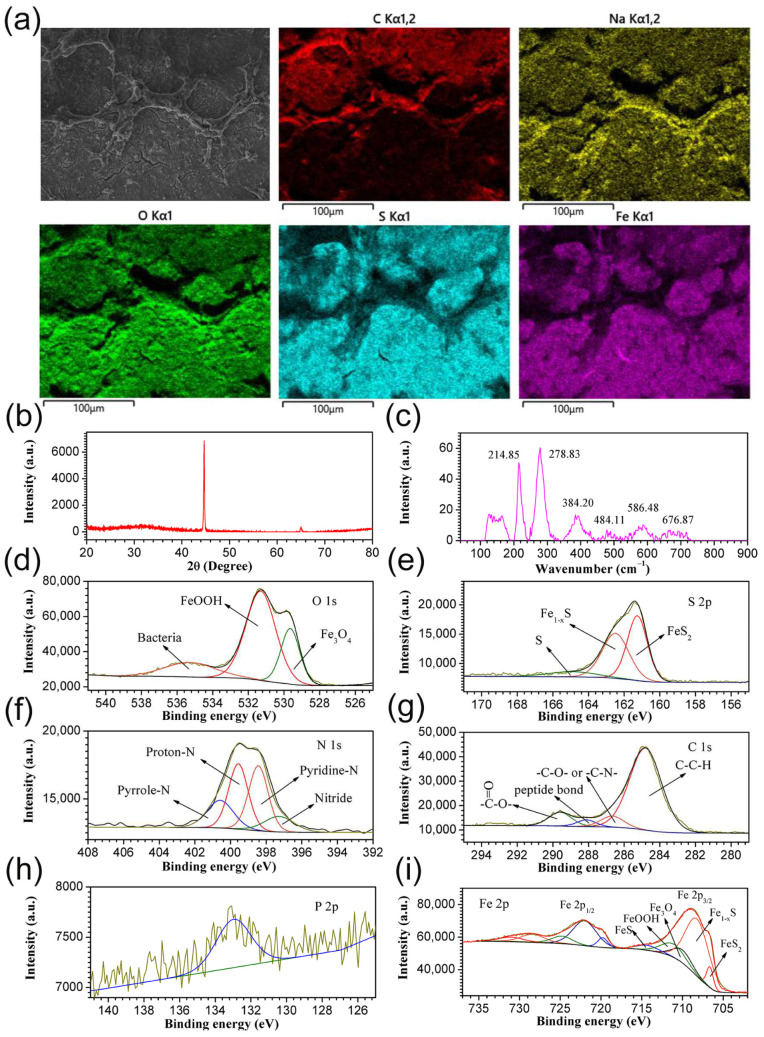
The chemical composition of the corrosion products. (**a**) EDS mapping images, (**b**) XRD patterns, (**c**) Raman spectrum and (**d**–**i**) high-resolution XPS spectra of O 1s, S 2p, N 1s, C 1s, P 2p and Fe 2p.

**Figure 5 ijerph-19-15416-f005:**
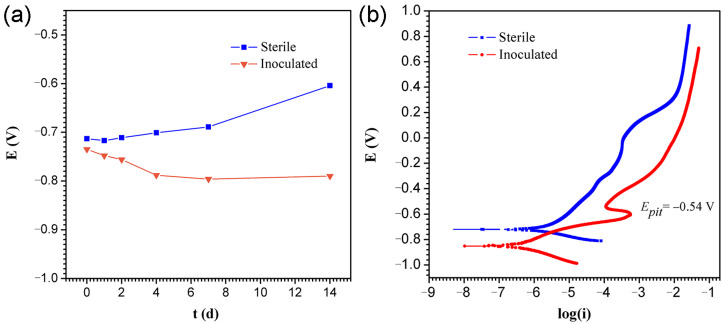
The value of OCP and the potentiodynamic polarization curves. (**a**) Variations of OCP vs. time during the 14-day incubation in sterile and PHS-Q-inoculated conditions and (**b**) the potentiodynamic polarization curves of the coupon in the sterile and PHS-Q-inoculated medium (scan rate: 1 mV s^−1^).

**Figure 6 ijerph-19-15416-f006:**
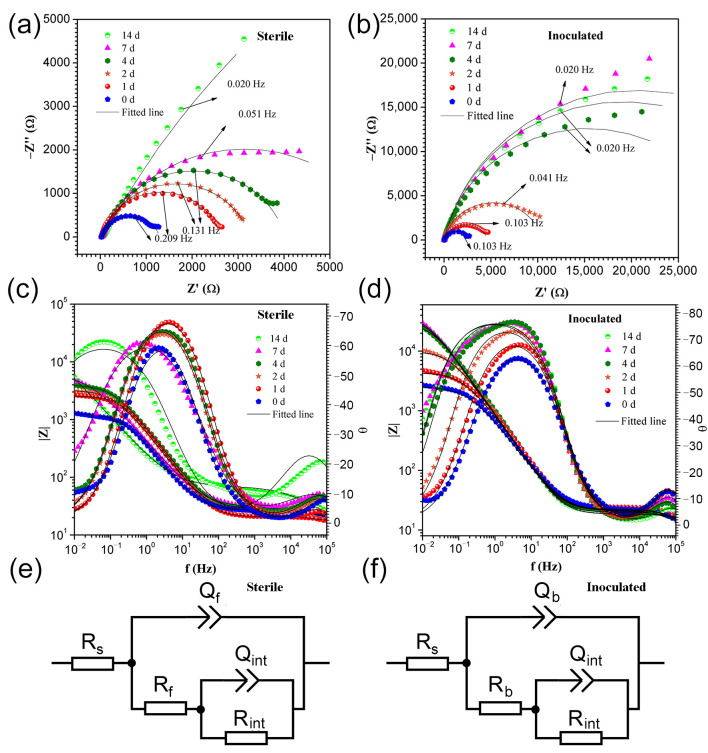
The electrochemical measurement results. The Nyquist and Bode plots for coupon in the (**a**,**c**) sterile medium and (**b**,**d**) PHS-Q-inoculated medium, and the equivalent circuit used for fitting the EIS (**e**,**f**).

**Figure 7 ijerph-19-15416-f007:**
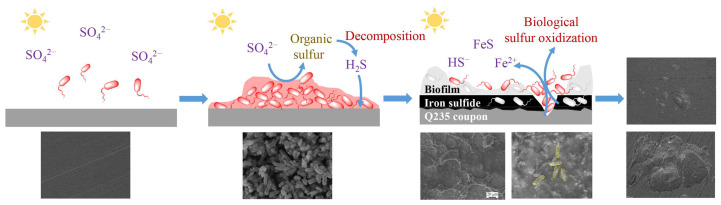
Schematic illustration of corrosion mechanism. The proposed corrosion processes of the Q235 carbon steel and the corresponding SEM images.

**Table 1 ijerph-19-15416-t001:** Fitted parameters of potentiodynamic polarization curves. The *β_a_*, *β_c_*, E*_corr_* and *I_corr_* value of samples in sterile and PHS-Q-inoculated condition, respectively.

	*β_a_* (mV dec^−1^)	*β_c_* (mV dec^−1^)	*E_corr_* (V vs. SCE)	*I_corr_* (10^−7^ A cm^−2^)
Sterile	154.28	−66.80	−0.72	17.20
PHS-Q-inoculated	130.52	−89.53	−0.85	4.46

**Table 2 ijerph-19-15416-t002:** Fitted parameters of EIS. EIS parameters of specimens in sterile and PHS-Q-inoculated conditions.

	Time (d)	*Q_f_* (MΩ^−1^ s^n^ cm^−2^)	n_f_	*R_f_* (Ω cm^2^)	*Q_int_* (kΩ^−1^ s^n^ cm^−2^)	n_int_	*R_int_* (kΩ cm^2^)
Sterile	0	1.52	0.87	7.57	0.57	0.81	1.27
	1	0.58	1.00	2.78	0.29	0.85	2.60
	2	1.91	0.88	7.86	0.35	0.82	3.23
	4	2.24	0.85	9.64	0.29	0.83	4.00
	7	34.32	0.63	14.56	0.58	0.74	6.22
	14	7.02	0.70	51.16	1.29	0.72	33.50
	Time (d)	*Q_b_* (MΩ**^−^**^1^ s^n^ cm**^−^**^2^)	n_1_	*R_b_* (Ω cm^2^)	*Q_int_* (kΩ^−1^ s^n^ cm^−2^)	n_2_	*R_int_* (kΩ cm^2^)
PHS-Q-inoculated	0	0.50	0.99	8.37	0.31	0.80	2.67
	1	0.41	0.99	8.77	0.24	0.84	4.61
	2	0.39	0.99	8.69	0.22	0.87	10.45
	4	0.63	0.99	5.61	0.21	0.87	30.78
	7	0.48	0.99	7.14	0.21	0.87	41.74
	14	0.59	0.99	3.88	0.22	0.87	38.30

## Data Availability

Not applicable.

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
