# Peer review of "Microbiologically Influenced Corrosion of Q235 Carbon Steel by Ectothiorhodospira sp."

_ijerph, 2022, doi:10.3390/ijerph192215416_

Round 1

Reviewer 1 Report

This paper presents results of the investigation of the influence of bacteria Ectothiorhodospira on the corrosion of carbon steel Q235. The researchers applied diverse techniques of surface analysis such as SEM, TEM, EDX, Raman, XPS and electrochemical techniques: EIS and polarization curves. The paper is well-structured, of high-quality presentation and of large relevance for the corrosion community. In my opinion, however, the manuscript should be still improved bevor it can be published.

Main concerns:

1.       3.3 Electrochemical analysis, lines 139-152. This paragraph describes the shift of the frequency of the phase peaks in the Bode-diagrams. I think, it is difficult to draw conclusions from this analysis. In my opinion, it should be more advantageous to start with the description of the equivalent circuit (line 152) and then discuss the variation of the different values of the electrical components.

Line 142: it would be more convenient to discuss in terms of concrete values and not of “low: … a low logarithmic impedance”: the term low needs from a reference. Also here, it should be written Impedance modulus. The term “logarithmic impedance” is not correct, the scale is logarithmic and not the impedance ?.

2.       Discussion, page 7, line 214: The physical meaning of the different circuit elements should be addressed: classically Qdl, and Rct refers to the CPE element and charge transfer resistance of the double layer. But in this case, they refer to the film-substrate interface! if you want to represent the iron dissolution reaction. The term double-layer is generally reserved for substrate-electrolyte interfaces. Perhaps better Qint, Rint? int: interfacial.

In the same way, Rf and Qf refer to the charge transfer and the CPE element of the film, respectively. This should be addressed in the discussion.

3.       Figure 5: in the case of corrosion in sterile medium, you do not have the biofilm, to which you ascribe the electrical elements Rf and Qf. If these elements are ascribed to the oxide film for the sterile experiment, indeed, it you be appropriately addressed!

4.       You used CPE elements for the description of the EIS results, but no data of the coefficient n are given! Also, apparently the results in figure 5 do not include the corresponding fitting curve, which would give indication of the fitting power of the model.

Minor corrections:

1.       Abstract, line 19: … oxidise FeS as the hydrogen donor… I think, much better: …. Use the oxidation of FeS as a hydrogen donor?

2.       In the abstract, it should be mentioned, that this occurs in localized areas.

3.       Page 1, line27: … natural environment is the presence of microbes.

4.       Page 1, line 37: …, by which metals…

5.       Page 1, line 40-41: I would write: Especially, facultative autotrophic SOB exist, which can oxidise sulfide to sulfate under photoautotrophic conditions to be used as the hydrogen donor and thus to fix carbon dioxide.

6.       Page 6, figure 5 e: scan rate?

7.       Page 7, line 216: the change of the corrosion potential does not necessarily mean an increase of corrosion rate, but rather a change of the corrosion mechanism, as it is here the case, due to the appearance of local areas with substrate/electrolyte interface.

8.       Page 8, line 225: … Ectothiorhodospira sp. induces pitting on metals…

9.       Page 8, line 234: In local areas, PHS-Q colonises directly on the ….

Author Response

Reviewer 1#:

This paper presents results of the investigation of the influence of bacteria Ectothiorhodospira on the corrosion of carbon steel Q235. The researchers applied diverse techniques of surface analysis such as SEM, TEM, EDX, Raman, XPS and electrochemical techniques: EIS and polarization curves. The paper is well-structured, of high-quality presentation and of large relevance for the corrosion community. In my opinion, however, the manuscript should be still improved bevor it can be published.

Main concerns:

Reviewer comment 1

3.3 Electrochemical analysis, lines 139-152. This paragraph describes the shift of the frequency of the phase peaks in the Bode-diagrams. I think, it is difficult to draw conclusions from this analysis. In my opinion, it should be more advantageous to start with the description of the equivalent circuit (line 152) and then discuss the variation of the different values of the electrical components.

Line 142: it would be more convenient to discuss in terms of concrete values and not of “low: … a low logarithmic impedance”: the term low needs from a reference. Also here, it should be written Impedance modulus. The term “logarithmic impedance” is not correct, the scale is logarithmic and not the impedance ?.

Response 1: Thank you very much for your kind and constructive comment. All of the authors strongly agree with your opinion and point of view. It is difficult to conclude from the original analysis, and we think the Bode diagram is difficult to analyse. But another reviewer asks us to analyse the Bode-diagrams in detail. Thus, we have to try our best to analyse them. If there are any problems, please point them out and give some suggestions. Thank you.

Moreover, we carefully describe the equivalent circuit (line 222-233) and then discuss the variation of the different values of the electrical components in the revised manuscript. We revised the description and used the concrete values in the sentences (line 200). We exchanged the term “logarithmic impedance” with impedance modulus. Thank you for your helpful comment.

Reviewer comment 2

Discussion, page 7, line 214: The physical meaning of the different circuit elements should be addressed: classically Qdl, and Rct refers to the CPE element and charge transfer resistance of the double layer. But in this case, they refer to the film-substrate interface! if you want to represent the iron dissolution reaction. The term double-layer is generally reserved for substrate-electrolyte interfaces. Perhaps better Qint, Rint? int: interfacial. In the same way, Rf and Qf refer to the charge transfer and the CPE element of the film, respectively. This should be addressed in the discussion.

Response 2: Thank you for your helpful comment. We agree with and adopt your suggestion. The CPE element and the charge transfer resistance in this work actually refer to the film-substrate interface.  We exchanged the Qdl and Rct by Qint and Rint in both the sterile and PHS-Q-inoculated group, as shown in Figure 6(e,f). Moreover, the meanings of Rf and Qf were addressed in the EIS results analysis section (line 226 ) per your suggestion.

Reviewer comment 3

Figure 5: in the case of corrosion in sterile medium, you do not have the biofilm, to which you ascribe the electrical elements Rf and Qf . If these elements are ascribed to the oxide film for the sterile experiment, indeed, it you be appropriately addressed!

Response 2: Thank you for your helpful comment. In order to clearly distinguish and describe the electrical elements in different experiment media, we name the resistance and capacitance of the biofilm/corrosion product film in the bacterium-inoculated medium as Rb and Qb, and name the resistance and capacitance of reaction product film in the sterile group as Rf and Qf.

Reviewer comment 4

You used CPE elements for the description of the EIS results, but no data of the coefficient n are given! Also, apparently the results in figure 5 do not include the corresponding fitting curve, which would give indication of the fitting power of the model.

Response 4: Thank you for your careful reading and helpful suggestion. The coefficient n and the fitting curve of the curves in the Nyquist plot were supplied in the revised manuscript, as shown in Table 2 and Figure 6, respectively.

Minor corrections: 

  1. Abstract, line 19: … oxidise FeS as the hydrogen donor… I think, much better: …. Use the oxidation of FeS as a hydrogen donor?
  2. In the abstract, it should be mentioned, that this occurs in localized areas.
  3. Page 1, line27: … natural environment is the presence ofmicrobes.
  4. Page 1, line 37: …, bywhich metals…
  5. Page 1, line 40-41: I would write: Especially, facultative autotrophic SOB exist, which can oxidise sulfide to sulfate under photoautotrophic conditions to be used as the hydrogen donor and thus to fix carbon dioxide.
  6. Page 6, figure 5 e: scan rate?
  7. Page 7, line 216: the change of the corrosion potential does not necessarily mean an increase of corrosion rate, but rather a change of the corrosion mechanism, as it is here the case, due to the appearance of local areas with substrate/electrolyte interface. 8.       Page 8, line 225: … Ectothiorhodospira spinducespitting on metals…
  8. Page 8, line 234: In local areas, PHS-Q colonisesdirectly on the ….

The minor corrections you pointed out whit other similar mistakes in the original manuscript were corrected. The scan rate of the potentiodynamic polarization curve was 1 mV s-1. The parameter was supplied in the “Materials and Methods” section in the revised manuscript. For the 7th problem you mentioned in the “Minor corrections” section, we realize that we probably over analysis the meaning of Ecorr. We revised it to a precise analysis: “The value of Ecorr was decreased in the PHS-Q-inoculated medium, suggesting the presence of PHS-Q changed the corrosion mechanism.” as you suggested. Convenient to review, all corrections have been highlighted with yellow in the revised manuscript. Thank you for your useful comment again.

Reviewer 2 Report

This manuscript deals with the microbiologically-influenced corrosion of Q235 carbon steel by a Gram negative, photosynthetic sulphur-oxidising bacterium, named Ectothiorhodospira sp. (PHS-Q). Microorganisms causing metal corrosion have been extensively investigated, however the effects of anaerobic iron-oxidising bacteria on the corrosion of metals like steel are still quite unknown. In this study, photoantotropic conditions have been used for corrosion experiments with the photosynthetic bacterium. Corrosion products were analysed mainly by SEM and XPS (in addition to other less informative characterization techniques like XRD, Raman spectroscopy, etc.) in order to evaluate the pitting corrosion on carbon steel induced by bacteria. Morphological investigation and electrochemical analysis proved that, owing to the bacteria ability to simultaneously oxidize/reduce sulphur-containing inorganic minerals, they are strongly corrosive for carbon steels.

The manuscript includes quite interesting results, useful to establish the relationship between biological sulphur cycle and iron corrosion. However, it is not clear which is the pH used for the cultivation medium (7.5?). This point is relevant for a corrosion study: metals like the investigated Q235 carbon steel sample are rapidly corroded by acids, and the presence of ammonium ions (NH4+) at high concentration (1g/liter) in the PHS-Q cultivation medium does not allow to establish with certainty what is the principal cause of the pitting observed on the sample surface. The experiment should be redesigned by modifying the cultivation medium composition (that is, the salt types), that should have a neutral pH. The other species contained in the cultivation medium (that is, HCO3- and HPO42-) are quite weak acids and therefore they do not influence significantly the corrosion process. According to Figure 2, corrosion takes place also in a sterile medium (the exposition time has not been indicated in the manuscript and it is required), therefore protons coming from the salt hydrolysis should have a role. Since the electrochemical tests have been performed in presence of bacteria, the observed differences in the Bode-phase for sterile and non-sterile conditions could be influenced by the bacteria presence. In general, biofilms protect the metallic substrate, thus preventing the corrosion, and the reason for the observed biofilm break down is difficult to justify. The presence of a biofilm on metal during incubation contrasts with corrosion enhancement. In addition, the reason for which the equivalent electric circuit predicts a protective effect of the biofilm, that is capable to decrease corrosion rate, is in accordance with this consideration.

Author Response

Reviewer 2#:

This manuscript deals with the microbiologically-influenced corrosion of Q235 carbon steel by a Gram negative, photosynthetic sulphur-oxidising bacterium, named Ectothiorhodospira sp. (PHS-Q). Microorganisms causing metal corrosion have been extensively investigated, however the effects of anaerobic iron-oxidising bacteria on the corrosion of metals like steel are still quite unknown. In this study, photoantotropic conditions have been used for corrosion experiments with the photosynthetic bacterium. Corrosion products were analysed mainly by SEM and XPS (in addition to other less informative characterization techniques like XRD, Raman spectroscopy, etc.) in order to evaluate the pitting corrosion on carbon steel induced by bacteria. Morphological investigation and electrochemical analysis proved that, owing to the bacteria ability to simultaneously oxidize/reduce sulphur-containing inorganic minerals, they are strongly corrosive for carbon steels.

The manuscript includes quite interesting results, useful to establish the relationship between biological sulphur cycle and iron corrosion. However, it is not clear which is the pH used for the cultivation medium (7.5?). This point is relevant for a corrosion study: metals like the investigated Q235 carbon steel sample are rapidly corroded by acids, and the presence of ammonium ions (NH4+) at high concentration (1g/liter) in the PHS-Q cultivation medium does not allow to establish with certainty what is the principal cause of the pitting observed on the sample surface. The experiment should be redesigned by modifying the cultivation medium composition (that is, the salt types), that should have a neutral pH. The other species contained in the cultivation medium (that is, HCO3- and HPO42-) are quite weak acids and therefore they do not influence significantly the corrosion process. According to Figure 2, corrosion takes place also in a sterile medium (the exposition time has not been indicated in the manuscript and it is required, therefore protons coming from the salt hydrolysis should have a role. Since the electrochemical tests have been performed in presence of bacteria, the observed differences in the Bode-phase for sterile and non-sterile conditions could be influenced by the bacteria presence. In general, biofilms protect the metallic substrate, thus preventing the corrosion, and the reason for the observed biofilm break down is difficult to justify.

The presence of a biofilm on metal during incubation contrasts with corrosion enhancement. In addition, the reason for which the equivalent electric circuit predicts a protective effect of the biofilm, that is capable to decrease corrosion rate, is in accordance with this consideration.

Response: Thank you very much for your kind comment and helpful suggestion. We are sorry that there were some missing details in the original manuscript. For example, the details of experimental methods were lost, including the information that the immersion corrosion test was conducted in the cultivation medium with pH 7.5 and that the immersion corrosion test period was 14 days in Figure 2. These pieces of missing information might be confusing and misleading. We have supplied the missing information in the revised manuscript.

The results of this work show that the Q235 specimen could be corroded by the sterile medium. However, none of the pitting corrosion was observed on the Q235 coupons in the thrice-repeated immersion corrosion test in the sterile medium. The corrosion of the Q235 coupon was uniform. The pitting corrosion just happened in the bacterium-inoculated medium. This conclusion was supported by the results of SEM and IFM. This result reveals that the ammonium ions (NH4+) affect the Q235 corrosion behavior but can not induce the pitting corrosion.

Moreover, to be exact, the presence of a biofilm on metal during incubation might contrast with the uniform corrosion enhancement. However, the pitting corrosion incurred by bacteria has other mechanisms. On the one hand, the corrosion products formed on the Q235 coupons in the sterile and PHS-Q-inoculated medium were different. The corrosion products formed in the PHS-Q-inoculated medium were mainly about FeSx. The corrosion product formed in a sterile medium was loose and easy to fall off. Its chemical composition can not be detected. But according to the composition of the cultivation medium, the corrosion product in the sterile medium could not be FeSx. That is to say, the appearance of PHS-Q incurs the formation of protective corrosion layer composed of biofilm and the corrosion products (FeSx) in a large area. On the other hand, we proposed that, in local areas, PHS-Q colonises directly on the iron sulfide surface. The sufficient sulfide and light initiated the pathway of photosynthetic autotrophic metabolism of PHS-Q. The sulfide formed on the surface near the PHS-Q-colonised area was gradually broken down, exposing the metal substrate in local areas. The exposed area of the Q235 coupon was very small compared with the protective corrosion layer area. As we know, the Icorr is the average corrosion current of the whole electrode plane. Thus, the value of corrosion current Icorr in a PHS-Q-inoculated medium was smaller than that in a sterile medium.

Reviewer 3 Report

The paper titled Microbiologically influenced corrosion of Q235 carbon steel by Ectothiorhodospira sp. By Hong Qi, Yingsi Wang, Jin Feng, Ruqun Peng, Qingshan Shi and Xiaobao Xie concerns with influence of autotrophic SOB on the pitting corrosion of carbon steel. Authors used different experimental methods for better understanding the influence of bacteria on corrosion mechanism. Achieved results were clearly showed with appropriate explanations. The sentences are well structured and convey the author's observations clearly and precisely. Figures are clear with adequate captions. However, manuscript needs to be improved in order to reach journal’s standard for publication.

Reviewer comment 1

In section, methods and materials, experimental procedure implemented in EIS and PP measurements need to be described.

Reviewer comment 2

The open circuit potential measurements may provide some information about corrosion mechanism of carbon steel with and without the presence of bacteria. In accordance with that, OCP measurements must be performed and achieved results need to be described in revised manuscript.

Reviewer comment 3

EIS and PP discussion of results is too poor. Explanation of the equivalent circuit used for fitting is also too poor. Authors need to explain what represent: Rs, Rf, Rct, Qf, Qdl. Besides those parameters, one of the most important parameter is n, that didn’t explain in the manuscript. Improve that.

Reviewer comment 4

Nyquist diagram explanation are too poor. The semicircle diameter changes with time exposure of metal to bacteria. Why? Explain that in revised manuscript.

Reviewer comment 5

Bode diagrams explanation are also too poor. The phase angle maximum changed with time. Explain that? Behavior of metal at higher and lower frequencies with and without the presence of the bacteria need to be explained.

Reviewer comment 6

After 14 days exposure of the metal to the sterile solution, the phase angle maximum is shifted in regard to the phase angle maximum reached in other experiments. Why? Explain that in revised manuscript.

Reviewer comment 7

The phase angle maximum at higher frequencies in sterile solution after 14 days reached higher values in regard to other experimental conditions. Explain that in revised manuscript.

Reviewer comment 8

In the presence of the bacteria, PP curves presented in Figure 5e show existence of anodic current peak at approximately -0.6 V. Explain.

Reviewer comment 9

All presented parameters changed with time in solution with and without bacteria. Authors need to discuss those changes in revised manuscript. Parameter n, also must be presented in Table 1 with adequate discussion.

Reviewer comment 10

The corrosion mechanism is determinate with the values of cathodic and anodic Tafel slopes. Discuss obtained values of ba and bc.      

Author Response

Reviewer 3#:

The paper titled Microbiologically influenced corrosion of Q235 carbon steel by Ectothiorhodospira sp. By Hong Qi, Yingsi Wang, Jin Feng, Ruqun Peng, Qingshan Shi and Xiaobao Xie concerns with influence of autotrophic SOB on the pitting corrosion of carbon steel. Authors used different experimental methods for better understanding the influence of bacteria on corrosion mechanism. Achieved results were clearly showed with appropriate explanations. The sentences are well structured and convey the author's observations clearly and precisely. Figures are clear with adequate captions. However, manuscript needs to be improved in order to reach journal’s standard for publication.

Reviewer comment 1

In section, methods and materials, experimental procedure implemented in EIS and PP measurements need to be described.

Response 1: Thank you very much for your helpful suggestion. The experimental procedure implemented in EIS and PP measurements has been added in the “Materials and Methods” section. The added content is as follows:

“2.4 Electrochemical measurements

The electrochemical test mainly about electrochemical impedance spectroscopy (EIS) and potentiodynamic polarization was performed on a Corrtest CS350 electrochemical workstation to characterize the electrochemical reactions and the formation of corrosion products and biofilms at the metal/biofilm interface. The electrochemical tests were performed in an anaerobic corrosion cell with a three electrodes system. The working electrode, reference electrode, and counter electrode were the Q235 steel, saturated calomel electrode, and platinum mesh, respectively. The cultivation medium was used as the electrolyte. The experiment group was inoculated with PHS-Q seed culture to provide an initial planktonic cell count of 106 cells mL−1. The open circuit potential (OCP) was steady after testing for about 1 h. EIS was performed based on the OCP, and the amplitude was set as 10 mV, the frequencies ranging from 100 kHz to 10 MHz. Potentiodynamic polarization curves were obtained at a scan rate of 1 mV s−1. The EIS of the system was tested on the initial, the 1st, 2nd, 4th, 7th, and 14th day, and the potentiodynamic polarization curves were tested on the 14th day. The data were analyzed by CS Studio 5. Each test was duplicated using three separate cells.”

Reviewer comment 2

The open circuit potential measurements may provide some information about corrosion mechanism of carbon steel with and without the presence of bacteria. In accordance with that, OCP measurements must be performed and achieved results need to be described in revised manuscript.

Response 2: Thank you for your comment. The OCP measurement on the initial, 1st, 2nd, 4th, 7th, and 14th day were performed in Figure 5(a). The corresponding results were described in the revised manuscript. The OCP of Q235 steel in a bacterium-inoculated medium was lower than that in a sterile medium. The results revealed that the OCP decrease could be attributed to an increase in the dissolution kinetics of the specimen surface caused by the PHS-Q.

Figure 5. The value of OCP and the potentiodynamic polarization curves. (a) Variations of OCP vs. time during the 14-day incubation in sterile and PHS-Q-inoculated conditions, (b) the poten-tiodynamic polarization curves of the coupon in the sterile and the PHS-Q inoculated medium (scan rate: 1 mV s−1).

Reviewer comment 3

EIS and PP discussion of results is too poor. Explanation of the equivalent circuit used for fitting is also too poor. Authors need to explain what represent: Rs, Rf, Rct, Qf, Qdl. Besides those parameters, one of the most important parameter is n, that didn’t explain in the manuscript. Improve that.

Response 3: Thank you for your constructive comment. The analyses of the EIS and PP results were expanded. The equivalent circuit used for fitting was explained in detail. For example, the meaning of the electrical elements was explained. Following the advice of the other reviewer’s suggestion, we renamed the electrical elements according to their meaning and the reaction process. In the equivalent circuit, Rs stand for the solution resistance. Rb and Qb stand for the resistance and capacitance of the biofilm/corrosion product film. Rf and Qf stand for the resistance and capacitance of the reaction product film in the sterile group. Rint and Qint stand for the charge transfer resistance and the capacitance of the film-substrate interface. The impedance of constant phase element (CPE) was used and shown below:

                                                 (1)

In which Y0 is the parameter related to the capacitance. Factor n is CPE power. It is an adjustable parameter that lies between 0 and 1. It can be used to measure surface inhomogeneity. The smaller the value of n, the larger roughness of the surface. Factor ω is the angular frequency.  Moreover, the values of n were displayed in Table 2.

Figure 6. The electrochemical measurement results. The Nyquist and Bode plots for coupon in the (a,c) sterile medium and (b,d) PHS-Q inoculated medium, and the equivalent circuit used for fitting the EIS (e,f).

 Table 2. Fitted parameters of EIS. EIS parameters of specimens in sterile and PHS-Q inoculated conditions.

time

(d)

Qf

( MΩ-1 sn cm-2)

n1

Rf

(Ω cm2)

Qint

( kΩ-1 sn cm-2)

n2

Rint

(kΩ cm2)

Sterile

0

1.52

0.87

7.57

0.57

0.81

1.27

1

0.58

1.00

2.78

0.29

0.85

2.60

2

1.91

0.88

7.86

0.35

0.82

3.23

4

2.24

0.85

9.64

0.29

0.83

4.00

7

34.32

0.63

14.56

0.58

0.74

6.22

14

7.02

0.70

51.16

1.29

0.72

33.50

time

(d)

Qb

( MΩ-1 sn cm-2)

n1

Rb

(Ω cm2)

Qint

( kΩ-1 sn cm-2)

n2

Rint

(kΩ cm2)

PHS-Q-inoculated

0

0.50

0.99

8.37

0.31

0.80

2.67

1

0.41

0.99

8.77

0.24

0.84

4.61

2

0.39

0.99

8.69

0.22

0.87

10.45

4

0.63

0.99

5.61

0.21

0.87

30.78

7

0.48

0.99

7.14

0.21

0.87

41.74

14

0.59

0.99

3.88

0.22

0.87

38.30

Reviewer comment 4

Nyquist diagram explanation are too poor. The semicircle diameter changes with time exposure of metal to bacteria. Why? Explain that in revised manuscript.

Response 4: Thank you for your constructive comment. We have expanded the analysis and the discussion content about the Nyquist diagram in the revised manuscript. Under PHS-Q-inoculated conditions, the Nyquist plot shows that the diameter of the impedance loop increases in the first 7 days and then decreases on the 14th day. This result reveals that a layer of the protective film gradually formed on the coupon surface in the PHS-Q-inoculated medium from the initial day and constructed a steady protective film on the 4th day of incubation. On the 14th day, the slight decrease in impedance modulus was induced by the localized breakdown of the protective film.

Reviewer comment 5

Bode diagrams explanation are also too poor. The phase angle maximum changed with time. Explain that? Behavior of metal at higher and lower frequencies with and without the presence of the bacteria need to be explained.

Response 5: Thank you for your constructive comment. We have added the explanation about Bode diagrams in the revised manuscript. The Bode diagrams of the sterile group reveal that the reaction resistance of the Q235 steel in the sterile medium increased with time. The shift of the maximum angle in the Bode plot might indicate the change in the charge transfer process. The Bode diagrams of the PHS-Q-inoculated group reveal that the impedance modulus values are larger than those in the sterile medium. There is a peak broaden but no peak shift during the 14 days incubation. These results indicate that a layer of the protective film gradually formed on the coupon surface in the PHS-Q-inoculated medium from the initial day and formed a steady protective film on the 4th day of incubation. On the 14th day, a slight decrease in impedance modulus was induced by the localized breakdown of the protective film. Moreover, the different impedance modulus values in sterile and PHS-Q-inoculated media reveal that the uniform corrosion rate of Q235 steel was slower in the PHS-Q-inoculated medium.

Reviewer comment 6

After 14 days exposure of the metal to the sterile solution, the phase angle maximum is shifted in regard to the phase angle maximum reached in other experiments. Why? Explain that in revised manuscript.

Response 6: Thank you for your constructive comment. We think the phase angle maximum was shifted to a lower frequency gradually. On the 7th day, the phase angle maximum has shown a significantly low-frequency shift tendency. With the time interval from the 7th day get to the 14th day, the shift looked much more obvious. The shift of the maximum angle in the Bode plot might indicate the change in the charge transfer process. In detail, the Bode phase angle plot shows that the maximum angle is observed initially in the frequency range of 1 to 10 Hz. For the remaining 14 days, the maximum angle first shifted to a high frequency from the 2nd to the 4th day. Then it shifted to a low frequency from the 7th day. Finally, it reached the 0.01 to 0.1 Hz range. The shift of the maximum angle in the Bode plot might indicate the cyclic formation and desorption of the reaction production film.

Reviewer comment 7

The phase angle maximum at higher frequencies in sterile solution after 14 days reached higher values in regard to other experimental conditions. Explain that in revised manuscript.

Response 7: Thank you for your comment. According to the variation tendency of the phase angle maximum at high frequency in regard to other experimental conditions, the abrupt phase angle shift on the 14th day can not be the instrument error. There must exist another factor that affects the electrode reaction process. However, we can’t find it out so far. The answer could be revealed in our future research.

Reviewer comment 8

In the presence of the bacteria, PP curves presented in Figure 5e show existence of anodic current peak at approximately -0.6 V. Explain.

Response 8: Thank you for your careful reading and helpful comment. In the PHS-Q-inoculated medium, the anodic current peak appears at approximately −0.54 V, indicating that the presence of PHS-Q could promote the passivation of the protection film. But PHS-Q fails to start the spontaneous passivation. The corresponding potential can be defined as pitting potential Epit.

Reviewer comment 9

All presented parameters changed with time in solution with and without bacteria. Authors need to discuss those changes in revised manuscript. Parameter n, also must be presented in Table 1 with adequate discussion.

Response 9: Thank you for your useful suggestion. We have expanded the discussion of the presented parameters, as follows:

Rf + Rint and Rb + Rint were closely related to the corrosion rate: The higher the value, the lower the corrosion rate. The data show that Rf and Rb are much smaller than Rint. Thus, the corrosion rate is decided by the value of Rint. The value of Rint in the PHS-Q-inoculated medium was greater than that in a sterile medium. The Rint data indicated that, in the PHS-Q-inoculated medium, the films were protective and decreased the uniform corrosion rate of metals. The Qb was less than Qf. According to the definition of a capacitor, the value of a capacitor is inversely related to its thickness. Thus, the biofilm/corrosion product film formed in the PHS-Q-inoculated medium is thicker than the reaction product film formed in the sterile medium. These results are consistent with the test results of potentiodynamic polarization and the analysis of the EIS plots.”

Reviewer comment 10

The corrosion mechanism is determinate with the values of cathodic and anodic Tafel slopes. Discuss obtained values of βa and βc.  

Response 10: Thank you for your helpful comment. We have added the discussion of the value of βa and βc in sterile and PHS-Q-inoculated medium. As follows:

“The Tafel slope is correlated with the rate of electrode reaction. A smaller value of the Tafel slope is indicative of easier electron transference [23]. For the coupon incubated in the PHS-Q-inoculated medium, the anode Tafel slope βa is smaller, and the cathodic Tafel slope βc is larger than that of the coupon incubated in a sterile medium, indicating that the kinetics of anode oxidation reactions were promoted. The PHS-Q made a detrimental effect on the Q235 or the protective film formed on the coupon surface, such as pitting corrosion. On the other hand, the cathodic reactions might be correlated with the development of  biofilm or corrosion products. The aforementioned results reveal that the Q235 carbon steel specimens undergo different corrosion processes in the presence and absence of the PHS-Q bacterium in the nutrient-rich medium.”

[23] Ramana, V. V.; Sasikala, C.; Ramaprasad, E. V. V.; Ramana, C. V. Description of Ectothiorhodospira salini sp. nov. J. Gen. Appl. Microbiol. 2010, 56, 313-319.

Round 2

Reviewer 1 Report

The authors  have properly revised the article according to the corrections advised by the referee.  Therefore, in my opinion, the paper can be now be published in the International Journal of Environmental Research and Public Health.

Reviewer 3 Report

The revised manuscript can be published in its present form.